# Design and In Vivo Pharmacokinetic Evaluation of Triamcinolone Acetonide Microcrystals-Loaded PLGA Microsphere for Increased Drug Retention in Knees after Intra-Articular Injection

**DOI:** 10.3390/pharmaceutics11080419

**Published:** 2019-08-19

**Authors:** Myoung Jin Ho, Hoe Taek Jeong, Sung Hyun Im, Hyung Tae Kim, Jeong Eun Lee, Jun Soo Park, Ha Ra Cho, Dong Yoon Kim, Young Wook Choi, Joon Woo Lee, Yong Seok Choi, Myung Joo Kang

**Affiliations:** 1College of Pharmacy, Dankook University, 119, Dandae-ro, Dongnam-gu, Cheonan-si, Chungcheongnam-do 31116, Korea; 2College of Pharmacy, Chung-Ang University, 84, Heukseok-ro, Dongjak-gu, Seoul 06974, Korea; 3Department of Radiology, Seoul National University Bundang Hospital, 82, Gumi-ro 173beon-gil, Bundang-gu, Seongnam-si, Gyeonggi-do 13620, Korea

**Keywords:** triamcinolone acetonide, microcrystal, PLGA microsphere, local delivery, spray-drying technique, intra-articular injection, joint retention, systemic exposure

## Abstract

A novel polymeric microsphere (MS) containing micronized triamcinolone acetonide (TA) in a crystalline state was structured to provide extended drug retention in joints after intra-articular (IA) injection. Microcrystals with a median diameter of 1.7 μm were prepared by ultra-sonication method, and incorporated into poly(lactic-*co*-glycolic acid)/poly(lactic acid) (PLGA/PLA) MSs using spray-drying technique. Cross-sectional observation and X-ray diffraction analysis showed that drug microcrystals were evenly embedded in the MSs, with a distinctive crystalline nature of TA. In vitro drug release from the novel MSs was markedly decelerated compared to those from the marketed crystalline suspension (Triam inj.^®^), or even 7.2 μm-sized TA crystals-loaded MSs. The novel system offered prolonged drug retention in rat joints, providing quantifiable TA remains over 28 days. Whereas, over 95% of IA TA was removed from joints within seven days, after injection of the marketed product. Systemic exposure of the steroidal compound was drastically decreased with the MSs, with <50% systemic exposure compared to that with the marketed product. The novel MS was physicochemically stable, with no changes in drug crystallinity and release profile over 12 months. Therefore, the TA microcrystals-loaded MS is expected to be beneficial in patients especially with osteoarthritis, with reduced IA dosing frequency.

## 1. Introduction

Intra-articular (IA) injection of corticosteroids, such as triamcinolone acetonide (TA) crystalline suspension, is commonly recommended to alleviate pain and inflammation in knee joints [1,2]. Marketed injectable TA suspensions are intended to be slowly dissolved in the synovial fluid, and the glucocorticoid molecules steadily bind to and activate the glucocorticoid receptors, obstructing the production of inflammation mediators, including prostaglandins, leukotrienes, and pro-inflammatory cytokines [3]. Nevertheless, the analgesic and/or anti-inflammatory effect of the TA crystalline suspension are reported to be weakened within two weeks following IA injection because of a rapid efflux of the drug from the arthritic joint [4,5,6]. As the synovial lining is ultra-structured with permeable intercellular gaps measuring 0.1–5.5 μm, therapeutic agents injected via the IA route tend to easily escape from the joint [7,8]. Moreover, the steroidal compound is quite soluble in aqueous media (21 μg/mL in phosphate buffered saline at 25 °C) [9], and TA crystals that have decreased to a few microns in size might be readily translocated into systemic circulation. In vitro release experiments showed that the marketed TA crystalline suspension was completely dissolved within 2 h [10].

Several pharmaceutical approaches, including hydrogels, liposomes, nanoparticles, and microparticles (MPs), have been explored to prolong the retention time of steroidal compounds in the synovial tissues, minimizing systemic exposure following IA injection [2,10,11,12,13,14,15]. When considering the leaky structure of the synovium, one of the sound strategies for localized delivery to the synovial tissues is administering the therapeutic agents to IA after entrapping in a micro-sized carrier system. Biocompatible and biodegradable polymeric MPs generally larger than 10 μm have been reported to be effective for remaining in the synovial cavity and providing a sustained-release profile in the joint [2,9,16]. Actually, a single IA injection of the PLGA MPs was clinically demonstrated to be effective in providing an extended retention time of the corticosteroid in joints while reducing the drug distribution in the bloodstream [17].

We previously formulated a PLGA MS system containing TA as in a stable crystalline form, to achieve the sustained-release profile in joints after IA injection [18]. Compared to the polymeric MS containing TA in an amorphous state, the TA crystals-loaded MSs prepared by layering the suspended drug crystals with PLGA polymer exhibited excellent physicochemical stability under storage condition (25 °C/60% R.H.) in terms of drug crystallinity and drug content in MSs. However, the MS system containing TA crystals with a median diameter of 7 μm could not effectively retard drug release, as the coating layer on the irregular TA crystals was unfair, with insufficient coating thickness; the extent of drug released from the MSs reached 75% for 12 h under sink condition. Thus, we assumed that the additional micronization process of the TA crystals might be beneficial, prior to encapsulation process, to obtain satisfactory coating thickness on the TA crystals, providing sustained-release profile in joints.

Herein, the goals of this study were to construct uniform TA microcrystals and embed the microcrystals in the polymeric MS, providing a prolonged retention profile in the joint following IA injection. The uniform drug microcrystals were prepared by an ultra-sonication method and then encapsulated into the polymeric MS by a spray-drying technique. The physicochemical characteristics of the microcrystals-loaded MSs were evaluated in terms of outer and inner structures, particle size, the drug loading amount, and loading efficiency. In vitro drug release patterns from the MSs were regulated by adjusting the ratio of PLGA to PLA polymers and the ratio of drug to the polymer. The in vivo concentration profile of TA in the plasma and joint tissue following IA injection of the microcrystals-loaded MSs were comparatively evaluated with those of the marketed TA crystalline suspension in rats.

## 2. Materials and Methods

### 2.1. Materials

TA powder and Triam inj.^®^ were kindly provided by Shinpoong Pharmaceutical Co. (Seoul, Korea). PLGA polymer with a lactide/glycolide ratio of 50:50 (5050DLG 4A, molecular weight 38,000–54,000 kDa) and PLA polymer (R203H, 18,000–24,000 kDa) were purchased from Lakeshore Biomaterials (Birmingham, AL, USA). Polysorbate 20, polysorbate 80, Sorbian monolaurate (Span 20), poly(ethyleneglycol) 4000 (PEG 4000), polyethylene-polypropylene glycol 188 (Poloxamer 188), cholesterol, benzalkonium chloride (BKC), sodium lauryl sulfate (SLS), and phosphate buffered saline tablets were purchased from Sigma Chemical Co. (St. Louis, MO, USA). Lecithin (l-α-phosphatidylcholine) was obtained from Avanti Polar Lipids, Inc. (Alabaster, AL, USA). Gelatin and glycerol were provided from TCI Chemicals, Co. (Tokyo, Japan). Acetonitrile (ACN), ethyl alcohol, and methanol of HPLC grade were obtained from J.T. Baker (Phillipsburg, NJ, USA). All other reagents were analytical grade.

### 2.2. Preparation of TA Microcrystals Using an Ultra-Sonication Method

Drug microcrystal suspension was fabricated using an ultra-sonication method previously reported with slight modifications [19]. Different kinds of stabilizers (polysorbate 20, polysorbate 80, span 20, PEG 4000, poloxamer 188, cholesterol, and BKC) were dissolved in ACN in the concentration ranges from 0.05 to 0.5 *w/v* % as shown is Table 1. TA powder (100 mg) was then added to the solution and vigorously vortexed for 5 min to disperse the drug powder homogeneously. The ultra-sonicator (Model Vibracell VC-505, Sonics and Materials Inc., Newtown, NC, USA) equipped with a 1/2-inch (13 mm) probe was placed into the TA suspension and it was sonicated for 3 min at 40% amplitude with 3 s pulses (on/off alteration). To prevent temperature elevation, the samples were located inside an ice bath during the sonication procedure. Prepared TA microcrystal suspensions were then stored at room temperature for further experiments.

### 2.3. Preparation of TA Microcrystals-Loaded MSs Using a Spray-Drying Technique

TA microcrystals-loaded MSs were fabricated by a spray-drying technique with a Buchi mini-spray dryer (Model B-290, Buchi Labortechnik AG, Flawil, Switzerland). The feeding solution was prepared by subsequently dissolving PLGA/PLA polymers and lecithin into the BKC-stabilized TA microcrystal suspension. The composition of each microcrystals-embedded MSs is represented in Table 2. The feeding solution was then pumped into the spray dryer nozzle at a feeding rate of 3 mL/min and a stirring rate of 250 rpm. The inlet and outlet temperatures were set to 70 °C and 45 °C, respectively, to evaporate the organic solvent. The atomizing air flow was 246 L/h and the aspirator capacity was 100%. Prepared microcrystals-loaded MS powders were collected and stored in a desiccator (Model OH-3S, As-one, Seoul, Korea) at 25 °C for 24 h to remove the residual organic solvent.

### 2.4. Morphological Features of TA Microcrystals and Microcrystals-Loaded MSs

#### 2.4.1. Appearance f TA Microcrystals and Microcrystals-Loaded MSs

Morphological features of raw material, TA microcrystals, and polymeric MSs were observed by SEM (Model Sigma 500, Carl Zeiss, Oberkochen, Germany). Drug powder and MSs samples were placed on a carbon tape and fixed onto an aluminum stub. The TA microcrystal suspension was dropwise loaded on the carbon tape and then dried for 6 h at room temperature to remove the aqueous vehicle. The platinum coating procedure was then conducted using an automatic sputter coater (Model 108Auto, Cressington, UK) at 15 mA. Appearance of samples was scrutinized by an electron microscope at an accelerated voltage of 15 kV.

#### 2.4.2. Cross-Sectional Image of TA Microcrystals-Loaded MSs

The internal structure of the microcrystals-loaded MSs was scrutinized by SEM after fixing the MSs in the gelatin blocks. At first, the gelatin medium was prepared by dissolving gelatin (20 *w/v* %) and glycerin (5 *w/v* %) in distilled water [20]. Approximately 40 mg of the MSs powder was dispersed in the 3 mL of gelatin medium inside the polystyrene disposable base mold (Tissue-Tek^®^, 15 × 15 × 5 mm) at 37 °C. The mold was then placed in a deep freezer maintained at −70 °C for 12 h. The frozen gelatin block was mounted to the cryostat stub (Model CM3050S, Leica Microsystems, Wetzlar, Germany) using an optimum cutting temperature compound (Sakura Finetechnical Co., Ltd., Tokyo, Japan). The MS-loaded block was then sectioned at a thickness of 20 µm at −20 °C and was immediately placed on the double-sided carbon tape. Samples were defrosted for 1 h at room temperature and coating and observation procedures were conducted using the same method as described above.

#### 2.4.3. Hyperspectral Mapping Images of TA Microcrystals-Loaded MSs

A hyperspectral microscopy imaging system (Model CytoViva^®^, Cytoviva Inc., Auburn, AL, USA) was employed to visualize TA microcrystals inside the MSs. The Cytoviva^®^ system included a BX-41 microscope (Olympus Corporation, Tokyo, Japan), a visible-near infrared hyperspectral imaging system, dual fluorescence module, and high-resolution adaptor. Approximately 10 μL of TA microcrystal suspension, blank MSs, and TA microcrystals-loaded MSs suspended in 1 mL of 0.5 *w/v* % polysorbate 80 solution was dropped onto a cover glass and the hyperspectral spectra were analyzed. The mapping process was performed on the TA microcrystals-loaded MS image, with acquired spectra of TA microcrystals and blank MSs (HyperVisual Software ENVI 4.8, ITT Visual Information Solutions, Boulder, CO, USA). The spectra corresponding to TA microcrystals and the blank MSs were expressed as red and yellow, respectively, in the hyperspectral image of TA microcrystals-loaded MSs.

### 2.5. Physicochemical Characterization TA Microcrystals and Microcrystals-Loaded MSs

#### 2.5.1. Crystallinity Analysis

The crystalline state of TA powder, TA microcrystals, blank MSs, and microcrystals-loaded MSs was analyzed using an X-ray diffractometer (XRD, Model Ultima IV, Rigaku, Tokyo, Japan) at 25 °C. For the TA microcrystal suspension, the aqueous vehicle was removed by centrifuging the suspension at 3500 g for 10 min, and subsequently, oven drying at 40 °C for 12 h. Each sample was put on the glass sample plate and the diffraction pattern over a 2*θ* range of 5–35° was determined using a step size of 0.02°. Voltage, current, and scan speed were set to 40 kV, 30 mA, and 1 s/step, respectively.

#### 2.5.2. Particle Size Analysis

Size distributions of TA microcrystals and MSs were determined by Mastersizer MS 2000 (Malvern Instruments Ltd., Worcestershire, UK) equipped with a Hydro 2000 S automatic dispersion unit. Prior to analysis, MSs powder was suspended in the aqueous medium consisting of 1 *w/v* % PEG 4000 and 0.5 *w/v* % polysorbate 20. The suspended samples were then dropwise added to an automatic dispersion unit to obtain a 10–15% range of obscuration. Sample and background measurement times were set to 5 s and 10 s, respectively, and 5 runs were conducted for each measurement. Mie theory was applied to calculate the size distribution by volume with the refractive index value of 1.52. The resultant particle sizes of the three batches were averaged and presented as mean ± standard deviation (SD) (*n* = 3). The d_0.5_, d_0.9_, and d_0.1_ indicated the median value defined as the diameter where 50%, 90%, and 10% of the population were below this value, respectively. SPAN value was an indicator representing the homogeneity of the particle size and was calculated by dividing the difference of d_0.9_ and d_0.1_ by d_0.5_ [21].

#### 2.5.3. Determination of Loading Amount and Efficiency of TA Microcrystals in MSs

To dissolve TA microcrystals-loaded MSs, 10 mg of MSs were added to 1 mL of dimethyl sulfoxide and then sonicated with a bath-type sonicator (Model 5510E-DTH, Bransonic, USA) for 10 min. The opaque solution was diluted with ACN and distilled water mixture (3:2 *v/v*) and was subsequently centrifuged at 16,000 *g* for 10 min to remove the precipitates. The concentration of TA in the supernatant was determined by a Waters HPLC system (Waters Corporation, Milford MA, USA) comprised of a pump (Model 515), auto sampler (Model 717 plus), UV detector (Model 486), and equipped with a Capcell Pak C18 column (150 mm × 2.0 mm, 3 μm, Shiseido, Tokyo, Japan). The mobile phase consisted of ACN and distilled water at a volume ratio of 3:2 and was eluted with a flow rate of 1.0 mL/min. The detection wavelength was set to 254 nm. The calibration curve of TA was linear in the concentration range of 1–100 μg/mL, with r^2^ values of 0.999. The drug loading amount and loading efficiency were calculated as follows [22]:
Drug loading amount = WL/WT,
(1)

Drug loading efficiency (%) = (WL/WF) × 100,
(2)
where W_L_, W_T_ and W_F_ represent the weight of TA in microcrystals-loaded MSs (mg), total weight of microcrystals-loaded MSs (mg) and feeding weight of TA (mg).

### 2.6. In Vitro Drug Release Profiles and Morphological Changes of Microcrystals-Loaded MSs

In vitro release profiles of TA from the novel MSs were comparatively evaluated with that of a marketed product under accelerated test conditions (45 ± 0.5 °C). To guarantee sink condition during the experiment, 0.5 *w/v* % of SLS and 0.05 *w/v* % of poloxamer 188, were added to 10 mM phosphate buffered saline (pH 7.4). MSs or the marketed product (Triam inj.^®^, TA 40 mg/mL) containing 20 mg of TA were immersed into 200 mL dissolution medium maintained at 45 ± 0.5 °C and then shaken with an agitation speed of 100 rpm. At predetermined intervals, 1 mL of the release medium was withdrawn and centrifuged at 16,000 *g* for 10 min. The supernatant was diluted two-fold with the mobile phase and TA concentration in the aliquot was determined by HPLC as described above. The equivalent volume of fresh pre-warmed dissolution medium was replenished to maintain a constant medium volume.

The morphological changes of the novel MSs during the in vitro release test were scrutinized by SEM. At predetermined intervals, MSs prepared with the PLGA:PLA ratio of 4:0 (F1), 1:3 (F4), and 0:4 (F5) were withdrawn and centrifuged at 900 *g*. The pelletized MSs were stored at −70 °C for 24 h and lyophilized for 24 h. The appearance of the lyophilized MS samples was observed by SEM with the same procedure described earlier.

### 2.7. In Vivo Systemic Exposure and Joint Retention of TA after IA Injection in Rats

#### 2.7.1. Animals and Experimental Protocols

In Vivo pharmacokinetic studies were performed after approval from the Institutional Animal Care and Use Committee (IACUC) of Seoul National University Bundang Hospital (approval number: BA1608-206/050-01, date of approval: August 9, 2016). Six-week-old male Sprague-Dawley rats (250 ± 20 g) were acquired from Samtako (Kyungki-do, Korea). Four or five rats were housed in each cage and kept in a temperature- and relative humidity-controlled room (23 ± 1 °C and 50 ± 5%, respectively) with a 12-h light-dark cycle. During the acclimatization period, rats were allowed free access to tap water and standardized chow.

After at least three days of the acclimatization period, rats were divided into three groups (*n* = 9 per group) by a stratified randomization scheme for similar body weights groups. The hair on both hind knee joints was removed using hair removal cream. Prior to IA injection, spray-dried MS (F4 and F8) were re-dispersed in the sterile diluent composed of 0.66 *w/v* % sodium chloride, 0.63 *w/v* % carboxymethylcellulose sodium, and 0.04 *w/v* % polysorbate 80 at the same drug concentration (2.5 mg/mL as TA). Each group received 50 μL of the marketed product, F4, and F8, respectively, using an insulin syringe (31 G) in both knee joints, to administer 125 µg of TA per knee. At the predetermined time, blood samples (approximately 0.2 mL) were collected from the submandibular vein using a 26 G heparinized syringe. Blood samples were centrifuged at 16,000 *g* for 10 min. The obtained plasma samples were then stored at −70 °C until being analyzed by LC-MS/MS assay.

Apart from the systemic exposure evaluation, knee samples were collected to estimate the level of TA in joint tissues. After 3, 7, 21, 28, and 42 days of the IA injection in both knees, two animals from each group were sacrificed, and both knees were removed using bone cutters. After removing any residual substances and adhered tissues, knees were accurately weighed and stored at −70 °C until LC-MS/MS analysis.

#### 2.7.2. LC-MS/MS Analysis of TA Concentration in Plasma and Knee Tissues

The TA concentrations in rat plasma or joint tissue were determined using the LC–MS/MS assay previously reported [23]. In brief, thawed plasma (100 μL) was mixed with 900 μL of methanol and vigorously vortexed for 10 min, to precipitate protein. After centrifuging at 16,000 g, a supernatant (10 μL) was analyzed through an LC-MS/MS system (Model LC-20 Prominence HPLC, Shimadzu and Model API 2000, AB/SCIEX, Foster City, CA, USA). In the case of articular samples, the frozen knee tissues were thawed and immersed in 2 mL of ACN and shaken overnight to extract TA from the tissue. The extracted solution was centrifuged at 16,000 *g* for 5 min and the supernatant was injected into the LC-MS/MS system. The transitions of 435.1/415.0/15 precursor ion (*m/z*)/product ion (*m/z*)/collision energy (V) were then monitored for TA. Data acquisition/analyses were conducted using Analyst^®^ version 1.5.2 software (ABSciex, Concord, ON, Canada). The assay was validated thoroughly and showed acceptable precision and accuracy, with a lower limit of quantification of 0.2 ng/mL in both rat plasma and knee tissue extract.

#### 2.7.3. Pharmacokinetic Parameters from TA Concentration Profile in Plasma

Pharmacokinetic parameters such as area under the plasma concentration versus time curve (AUC_0–7days_), maximum plasma concentration (C_max_), time needed to reach the maximum plasma concentration (T_max_), and terminal half-life (T_1/2_) in plasma were calculated using the linear trapezoidal rule in the BA Calc 2007 pharmacokinetic analysis program (Korea Food & Drug Administration, Seoul, Korea).

### 2.8. Physicochemical Stability of TA Microcrystals-Loaded MSs

The long-term storage stability of the novel MSs was evaluated in terms of drug crystallinity, drug content, and in vitro release profile. TA microcrystals-loaded MS (F4) power was placed into the scintillation vial and was stored in the chamber maintained at 25 °C and 60% R.H. After 12 months of storage, the drug crystallinity, drug content, and in vitro release profile were evaluated with the same method as previously described.

### 2.9. Statistical Analysis

Each experiment was performed at least thrice and the data are presented as the mean ± SD. Statistical significance was determined using a one-way analysis of variance (ANOVA) test and was considered to be significant at *p* < 0.05 unless otherwise indicated.

## 3. Results and Discussion

### 3.1. Formulation and Physical Characteristics of TA Microcrystals

Various stabilizers were screened to micronize TA powder in the organic solvent using a probe type ultra-sonicator (Table 1). ACN was employed as the vehicle as it exhibited low solubility for the steroidal compound (<1 mg/mL), and high solvation capacity for PLGA and PLA polymers [18]. When the surface stabilizer was not included in the vehicle, TA powder was not uniformly dispersed in the medium after the homogenization process, rather forming large precipitates. The addition of steric stabilizers, such as polysorbate 20, polysorbate 80, PEG 4000, poloxamer 188, and cholesterol could not provide a uniform dispersion of the split TA microcrystals, forming drug aggregates within 24 h. On the other hand, when BKC was included in the organic vehicle at a concentration of 0.1 to 0.5 *w/v* %, TA microcrystals with a median size below 2.1 μm were shaped with re-dispersibility in the organic solvent (Table 1 and Figure 1A). When the concentration of the cationic surfactant was less than 0.05 *w/v* %, it could not afford the re-dispersibility of TA microcrystals. Thus, BKC at the concentration of 0.1 *w/v* % was employed for further preparation of TA microcrystal suspension in ACN.

The morphological feature of the TA microcrystals stabilized by 0.1 *w/v* % BKC was observed by FE-SEM. TA raw material showed characteristic crystal forms, such as hexahedron, octahedron, and dodecahedrons, with different sizes in the range from 2 to 20 μm (Figure 1C). Whereas, the crystal size was markedly decreased to 1–3 μm by the ultra-sonication process (Figure 1D), coinciding with the crystal size as determined by Mastersizer (d_0.5_, 1.7 μm). In spite of crystal size reduction, no noticeable change to the shape was observed in the TA microcrystals. The crystalline state of TA microcrystals was further evaluated by comparing the X-ray diffraction spectrum of TA microcrystals with that of the raw material (Figure 1B). The spectrum of TA microcrystals was identical to that of drug powder, exhibiting distinctive diffraction peaks at 2*θ* equal to 9.9°, 14.5°, 17.6°, and 24.7°. On the other hand, the cationic surfactant showed no distinctive diffraction peaks over the 2*θ* range of 5–35°. Taken together, we concluded that TA powder was effectively micronized to 1–3 μm, with no crystalline changes during the ultra-sonication process.

### 3.2. Formulation and Physical Characteristics of TA Microcrystals-Loaded MSs

Various TA microcrystals (1.7 μm median size) or intact TA powder (7.2 μm)-embedded MSs were fabricated using the spray-drying technique, and these particle size, homogeneity, drug loading amount, and loading efficiency are represented in Table 2. The median particle size of MSs prepared ranged from 15.8 to 18.8 μm, with a narrow size distribution possessing a SPAN value below 2.1. The formulation variables, such as TA crystal size, the ratio of PLGA and PLA polymers, and the ratio of drug to polymer, did not cause marked differences in size and homogeneity of TA-loaded MSs (Table 2). The particle size of the novel MS was considered to be suitable for IA prolonged delivery, preventing the trans-synovial efflux of injected TA microcrystals [17,24,25]. There was also no remarkable difference in the drug loading efficiency in the polymeric MSs between the formulas, exhibiting more than 90% drug loading efficiency in all formulations. The absence of the external phase during the spray-drying process might prevent distribution and/or diffusion of TA microcrystals during the external phase, and thus promote TA crystals to be located in the polymeric matrix after solvent evaporation, irrespective of composition variables. On the other hand, the loading amount of TA in MS was adjusted from 0.09 to 0.31 *w/w*, by controlling the drug to polymer weight ratio from 1:2 to 1:10.

The novel TAs-loaded MSs were further characterized in terms of outer and internal structures and drug crystallinity in MS (Figure 2). The MSs (F4) prepared by the spray-drying technique was highly spherical, with a smooth and homogeneous surface (Figure 2A). In the cross-sectional image, the microcrystals showed different textures from the polymeric matrix and were found to be uniformly embedded in the polymeric matrix (Figure 2B). The number of microcrystals loaded per MS was elucidated by translating the loading amount into the number of MS and TA microcrystals. In the process of converting the weight to a number, the volume of single MS and TA microcrystal was calculated with the assumption that the shape of the MS and microcrystal were spherical and cubic, respectively, with both having a density of 1.0. In the MS formulations prepared with the drug to polymer ratio of 1:2 (F6), 1:3 (F7), 1:5 (F4), and 1:10 (F8), the number of microcrystals embedded in each MS was calculated as 215, 94, 82, and 50, respectively. In the hyperspectral image (Figure 2C), TA microcrystals (red color) were observed to be principally located inside the PLGA/PLA MS (yellow color). However, individual microcrystals were not separately spotted in the image, probably because of the low resolution of Cytoviva^®^. The characteristic peak of TA microcrystals was identically detected in the microcrystals-load MS (Figure 2D), denoting that TA microcrystals stabilized by BKC were successfully incorporated in the MSs, with no crystalline changes during the fabrication process.

### 3.3. In Vitro Drug Release and Degradation Profiles of TA Microcrystals-Loaded MSs

In Vitro drug release profiles from the marketed product, the drug powder- or micronized TA crystals-loaded MSs were evaluated under accelerated dissolution conditions. Although the synovial fluid does not assure the sink condition for the drug dissolution, the in vitro release test under sink conditions was favored for quicker comparison between the formulations. Moreover, as the drug release profiles from PLGA/PLA MSs could be retarded from days to months at body temperature (37 °C), the liberation pattern of the steroidal compound from the MSs was further facilitated by elevating the temperature of the dissolution media (45 °C), promoting the degradation and/or hydrolysis of the biodegradable polymers [26,27,28]. Actually, the accelerated test at high temperature was reported to be beneficial for faster comparison of release behavior between MS formulas, with high correlation with that obtained at 37 °C [27]. Shen and Burgess (2012) revealed that the time required to reach 100% drug release from the MSs prepared with PLGA polymer with the glass transition temperature (T_g_) of 44–48 °C was determined to be 10, 5, 3, and 1.3 days, respectively, at temperatures of 45, 50, 53, and 60 °C. Herein, the temperature of the dissolution medium was set to 45 °C, which did not exceed the T_g_ values of both polymers (46–52 °C). When the polymeric MSs were exposed to the medium at temperatures above the T_g_ of the polymer, the drug diffusion coefficient was drastically increased [29], diminishing the difference between the release profiles between the polymeric particulates.

Under the accelerated condition, the marketed product containing 13 μm-sized TA crystals stabilized by polysorbate 20 and sodium carboxymethyl cellulose was rapidly liquefied in the aqueous medium, showing complete drug release within 90 min (Figure 3A). It coincided with a previous report that showed that the TA crystal suspension was readily dissolved in phosphate buffered saline within 2 h [10]. The drug release from the MS with 7.2-μm-sized TA crystals (F0) was not markedly retarded compared to the intact drug crystal, releasing over 95% of TA within 3 h. The incomplete and/or erratic coating thickness of the polymeric layer on the TA crystals probably could not effectively restrain the dissolution and diffusion procedures of the TA crystals into the aqueous media. On the other hand, drug release from the MSs containing smaller TA crystals (median size of 1.7 μm) was markedly impeded compared to the marketed product or 7.2-μm-sized TA crystal-loaded MS, especially as the ratio of PLA increased in the MS (Figure 3A).

As the glycolic acid has faster hydration/swelling behavior compared to lactic acid [30], MSs prepared with over 50% of PLGA polymers (F1, F2, and F3) showed higher burst release, with over 60% of drug released within 2 days. After the initial burst release, the extent of TA liberated from the F1, F2, and F3 polymers continuously rose, exhibiting over 90% release after 7 days under sink condition. On the other hand, the F4 formula with a PLGA:PLA ratio of 1:3, exhibited a more protracted release profile compared to that of F1, F2, and F3, exhibiting a linear release pattern for 21 days after a 53% initial release in the first 2 days. F5 (PLGA:PLA ratio of 0:4) showed the slowest release profile, displaying only 52% of the accumulated drug release after 21 days. Although there was marked difference in release profile depending on the ratio of PLGA and PLA polymer, the drug release pattern from novel drug microcrystals-loaded MSs were characterized by initial burst release and subsequent slow release profile, which is consistent with the typical release pattern of PLGA/PLA based MPs previous reported [31,32]. In the early phase, TA microcrystals located on or inner compartment near the surface of the MS might be rapidly dissolved by surrounding and/or penetrated aqueous media, and released from polymeric matrix mainly by diffusion mechanism. After initial burst release, the drug release rate tended to be retarded, due to the increased diffusion distance. Afterward, and the remaining steroidal compound in the MSs might be liberated by polymeric degradation and erosion and/or collapse of polymeric MSs.

Different drug release patterns depended on the PLGA:PLA ratio were highly consistent by the morphological changes of TA microcrystals-loaded MSs. As shown in Figure 4, because of the rapid swelling and hydrolysis nature of PLGA polymer, the PLGA MSs (F1) began to collapse and was excavated within three days. Thus, the drug microcrystals embedded in the PLGA MS might be readily exposed to aqueous media, and immediately dissolved under sink condition. On the contrary, because of the greater hydrophobicity of the PLA polymer compared to the PLGA polymer, the hydrolytic degradation of PLA MS (F5, PLGA:PLA = 0:4) progressed slowly. When the MSs were scrutinized at 3 and 7 days, fine pores were formed on the roughed surface and the pore size was gradually enlarged as time elapsed. Nevertheless, the overall globular shape and dimension of MS were retained even at 21 days, supporting the slow and incomplete release profile of TA from the MS (F7). The degradation pattern of F4 prepared with the PLGA:PLA ratio of 1:3 was intermediate between those of MSs prepared with PLGA or PLA polymer F1 and F5, respectively. After surface erosion and pore formation at three days, the MS was then gradually collapsed over 21 days. The drug release rate from F4 was markedly delayed compared to that from F1 but was much faster and higher than that from F5, releasing over 80% of the drug loaded for 21 days. The ratio of PLGA to PLA polymers was fixed to 1:3, expecting the prolonged release pattern for further investigation.

The in vitro release profile of TA from the PLGA/PLA MSs prepared with different drug to polymer ratios was further evaluated. As shown in Figure 3B, the initial drug release from the novel MSs were gradually decreased, as the drug to polymer ratio was increased. When the ratio of drug to polymer was 1:2 (F6) or 1:3 (F7), the percentage of drug released for 24 h had reached approximately 86% and 78%, respectively. On the other hand, in the formulations of the drugs:polymer ratio of 1:5 (F4) and 1:10 (F8), the release of the steroidal compound from the MSs were markedly retarded, exhibiting less than 60% of drug release over 5 days. Drug release from the F4 or F8 was steeped after 7 days, probably because of erosion and/or collapse of the MSs, but prolonged for 21 days. Formulas F4 and F8 were further exploited for in vivo pharmacokinetic study in rats, expecting an extended release profile over one month in the knee joint.

### 3.4. In Vivo Systemic Exposure and Ioint Retention of TA after IA Injection in Rats

The systemic exposure and local bioavailability of TA following a single IA injection of the marketed product or the novel MSs (F4 and F8) were evaluated in normal rats. The IA dose of TA treated in all groups was same to 0.25 mg per knee, which was well tolerated in rats [10]. The plasma levels of TA as a function of time following IA injection of the marketed product, F4, and F8 are represented in Figure 5 and the relevant PK parameters are summarized in Table 3. It is recommended that the exposure of the steroidal compound in blood be minimized, as the exogenous corticosteroid can cause Cushing syndrome, incurred impaired wound healing, infection, and muscle weakness [10,33,34]. However, unfortunately, the plasma level of TA was drastically elevated after administration of the marketed product, reaching C_max_ value of 218.7 ng/mL after 3.7 h. This rapid redistribution of TA into the bloodstream is in agreement with earlier reports that intra-articularly injected TA crystals were rapidly absorbed, with a T_max_ value of 4 h in patients with osteoarthritis [6,35,36]. This rapid drug efflux from the knee joint is also correlated with in vitro release profiles, denoting that TA crystalline suspension injected in the joint might be rapidly dissolved and passed out the gap in the synovial membrane. After reaching a C_max_ of 3.7 h, the plasma level of TA sharply decreased below 30 ng/mL after 12 h post-administration of the marketed product.

In contrast, the systemic exposure of the exogenous corticosteroid was markedly decreased following IA injection of the TA microcrystals-loaded MSs (F4 and F8). The C_max_ values of TA following IA injection of F4 and F8 were determined to 75.6 ng/mL and 32.2 ng/mL, respectively, which were only 34% (*p* < 0.05) and 15% (*p* < 0.05) that of the marketed product. Correspondingly, AUC_0–7days_ values in the F4- and F8-treated groups were drastically decreased to less than 54% (*p* < 0.05) and 37% (*p* < 0.05) of that obtained from the marketed product, respectively. These pharmacokinetic data indicated that the novel MSs remarkably lessened the redistribution of dissolved and/or micronized compound into the bloodstream, prolonging the retention time of TA in the knee. Between the two groups treated with MSs prepared with loading amount of 0.16 (F4) and 0.09 (F8), respectively, the drug exposure to blood was much lowered in the F8-treated group, showing 68% and 42% decreased AUC_0-7days_ and C_max_ values compared to those obtained from the F4-treated group. This pharmacokinetic tendency is explainable with the in vitro release test results, which revealed that the extent of drug released from the polymeric matrix declined as the drug to polymer ratio increased.

The drug remaining in the joint tissue following IA single injection of each formula was further assessed in normal rats (Figure 6). After the IA injection of the marketed product, the drug concentration in the joint at three days post-dosing was only 5.6 μg/g because TA crystals were quickly effluxed from the joint tissue. The percentages of the drug remaining in the joint tissue at 3 and 7 days were calculated to be only 4.5% and 2.4%, respectively. After 21 days, the drug concentration in the joint tissue was below the limit of detection. This result is in line with a previous report that only two of eight patients with osteoarthritis had quantifiable synovial TA concentration at week 6, following IA injection of the marketed product [37]. On the other hand, the novel MS formulations exhibited a markedly profound and prolonged concentration profile in joint tissue compared to the marketed product, exhibiting quantifiable TA concentration over 28 days. Three days after the single administration of F4 or F8, the drug concentration in joints was determined to be 45 μg/g and 67 μg/g, respectively, which is one-third and one-half of the initial dose. In both MSs-treated groups, the TA concentration in the tissue gradually decreased as time elapsed, but approximately 5% of the initial dose was still detected at 28 days. These findings suggested that the retention time of TA in the joint tissue was extended with the sustained-release pattern of the novel MSs.

### 3.5. Long-Term Stability of TA Microcrystals-Loaded MS

The physicochemical stability of the novel MS was evaluated after 12 weeks of storage under ambient conditions (25 °C, 60% RH). The storage condition of the MSs was set to ambient condition, as there was morphological change above 40 °C, due to softening of the polymer over T_g_. At first, the drug crystallinity in the MS was assessed using XRD because a change in drug crystalline nature may occur during storage, affecting the drug chemical stability and release pattern from the MS. Under ambient conditions, the crystallinity of TA microcrystals embedded in the MS was stably maintained over 12 months, with no changes in diffraction pattern (Figure A1). There was also no change in drug content in F8 MS, displaying over 97% drug content after 12 months of storage. The in vitro dissolution pattern was also comparable with that of MSs immediately prepared, exhibiting a sustained-release profile of over 21 days (Figure A1). From these findings, we concluded that the novel MS system was physicochemically stable at least for one year under ambient conditions.

## 4. Conclusions

A novel parenteral sustained-release system of TA was successfully prepared by micronizing TA powder into 1.7 μm-sized microcrystals, and subsequently embedding into PLGA/PGA polymeric MSs using a spray-drying technique. TA microcrystals were efficiently entrapped into the polymeric MSs, preserving their distinctive crystalline nature. In vitro drug release from the novel MSs was markedly retarded compared to the marketed product and even 7.2 μm-sized TA crystal-loaded MSs, exhibiting a prolonged release profile over 21 days under accelerated conditions (45 °C). In an in vivo pharmacokinetic study in normal rats, the duration that the TA remained in the joint tissue was markedly extended, providing profound drug remains at 28 days following IA single injection. Moreover, TA microcrystals-loaded MSs drastically decreased the systemic exposure of the steroidal compound compared to the marketed product. Thus, the novel IA long-acting system could be a valuable tool, providing both increased drug retention in the knee and diminished systemic exposure of TA following a single administration.

## Figures and Tables

**Figure 1 pharmaceutics-11-00419-f001:**
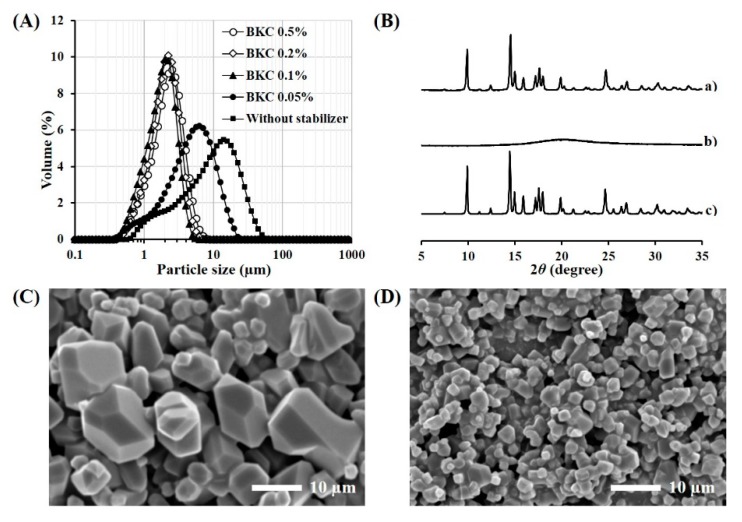
Morphological and physical characteristics of TA microcrystals. (**A**) Size distribution of the drug microcrystals stabilized by BKC, (**B**) XRD patterns of (a) TA raw material, (b) BKC powder, and (c) TA microcrystals, SEM images of (**C**) TA raw material, and (**D**) TA microcrystals stabilized by BKC at the concentration of 0.1 *w/v* %.

**Figure 2 pharmaceutics-11-00419-f002:**
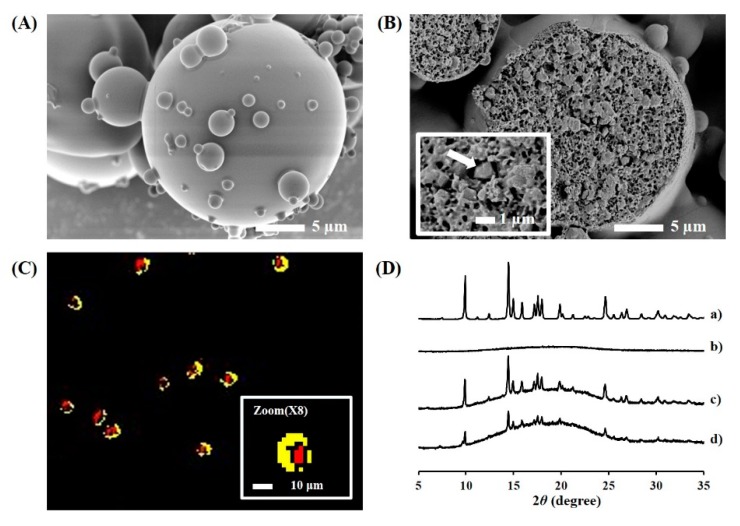
Morphological and physical characteristics of the microcrystals-loaded MSs. Representative micrograph of (**A**) intact and (**B**) cross-sectioned microcrystals-loaded MS (F4), (**C**) hyperspectral image of microcrystals-loaded MSs, and (**D**) XRD patterns of (a) TA microcrystals, (b) blank MS, (c) F4 MS (drug:polymer = 1:5), and (d) F8 MS (1:10); Notes: Inset in (**B**) is ×5000 magnified images and the arrow points to the TA crystal surrounded by the polymeric matrices. In the hyperspectral image (**C**), PLGA/PLA polymers and TA microcrystals are colored as yellow and red, respectively.

**Figure 3 pharmaceutics-11-00419-f003:**
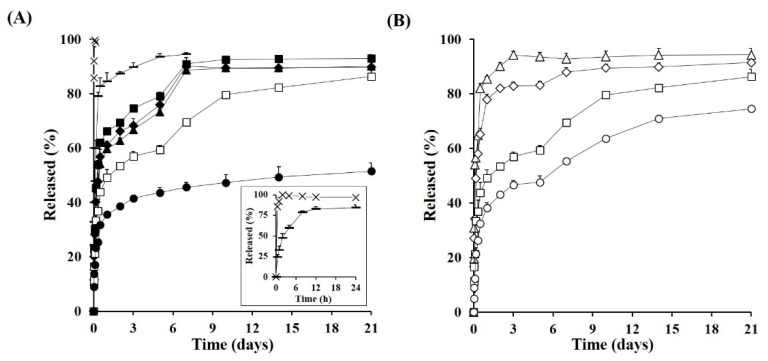
In Vitro release profile of TA from the novel MSs under accelerated conditions (45 °C). (**A**) Drug release profile from the marketed product (×), 7.2 μm-sized TA crystals-loaded MS (F0, –), and 1.7 μm-sized TA crystals-loaded MS prepared with different PLGA:PLA ratios; 4:0 (F1, ■), 3:1 (F2, ♦), 2:2 (F3, ▲), 1:3 (F4, □), and 0:4 (F5, ●) and (**B**) drug release profile from MSs prepared with different drug:polymer ratios; 1:2 (F6, △), 1:3 (F7, ◇), 1:5 (F4, □), and 1:10 (F8, ○); Notes: The inset graph (**A**) is the magnified release profile from the marketed product and 7.2 μm-sized TA crystals-loaded MS. Data are expressed as the mean value (*n* = 3) and error bars are SDs.

**Figure 4 pharmaceutics-11-00419-f004:**
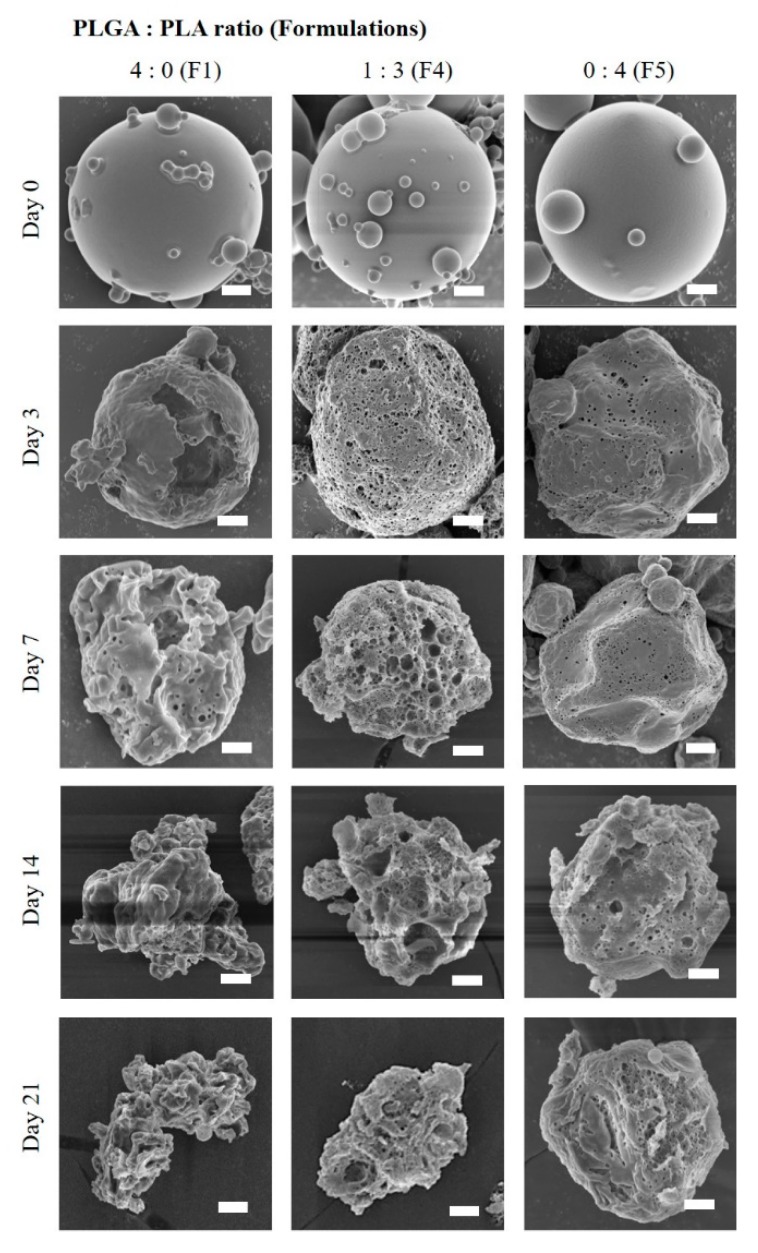
Morphological changes of TA microcrystals-loaded MSs prepared with different PLGA:PLA ratios, 0:4 (F1), 1:3 (F4), 0:4 (F5) under the accelerated release conditions (45 °C); Note: Scale bars in each image indicate 2.0 μm.

**Figure 5 pharmaceutics-11-00419-f005:**
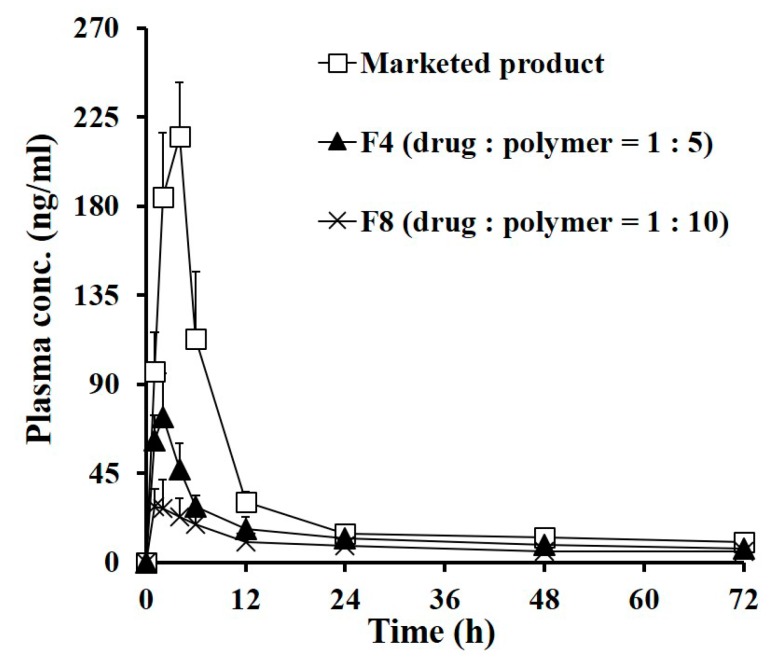
Plasma concentration–time profiles of TA following IA administration of the marketed product, F4 (drug:polymer = 1:5) and F8 (drug:polymer = 1:10) in rats at a dose of 0.5 mg/kg; Note: Each point represents mean ± SD (*n* = 5).

**Figure 6 pharmaceutics-11-00419-f006:**
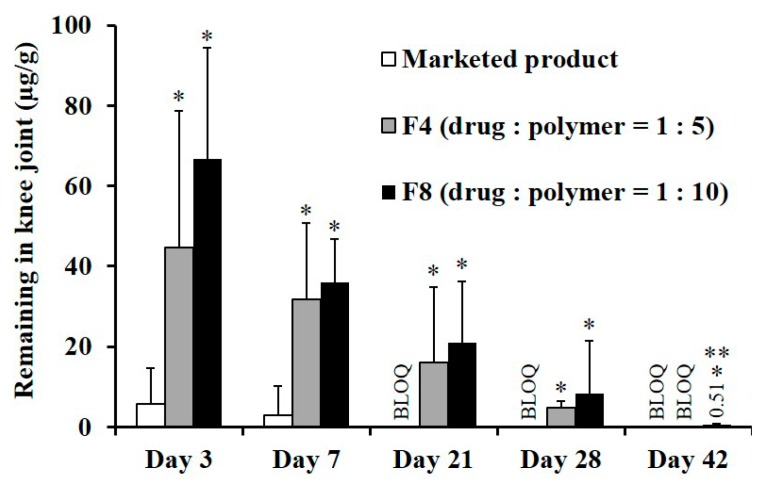
TA remains in rat joint tissue following IA administration of the marketed product, F4, and F8 at a dose of 125 μg of TA per knee; Notes: Vertical bars represent mean ± SD (*n* = 6). Statistical analysis was performed using the one-way ANOVA test; * significantly different from the marketed product (*p* < 0.05), ** significantly different from F4 (*p* < 0.05). BLOQ value of LC-MS/MS analysis was determined to 0.2 ng/mL.

**Table 1 pharmaceutics-11-00419-t001:** Effects of kinds of stabilizers on size, homogeneity, and dispersibility of TA microcrystals in ACN.

Stabilizer (*w/v* %) ^1^	Crystal Size (d_0.5_, µm) ^2,3^	Homogeneity (SPAN) ^2,4^	Dispersibility ^5^
- ^6^	7.21 ± 1.02	2.14 ± 0.13	Aggregated
Polysorbate 20 0.5%	6.63 ± 0.82	2.11 ± 0.07	Aggregated
Polysorbate 80 0.5%	7.35 ± 0.91	1.99 ± 0.04	Aggregated
Span 20 0.5%	17.7 ± 6.27	1.81 ± 0.11	Aggregated
PEG 4000 0.5%	7.15 ± 3.07	2.10 ± 0.88	Aggregated
Poloxamer 188 0.5%	5.44 ± 1.84	2.00 ± 0.52	Aggregated
Cholesterol 0.5%	8.72 ± 3.49	1.81 ± 0.60	Aggregated
BKC 0.5%	2.11 ± 0.05	1.32 ± 0.03	Re-dispersible
BKC 0.2%	1.94 ± 0.06	1.20 ± 0.03	Re-dispersible
BKC 0.1%	1.73 ± 0.02	1.21 ± 0.01	Re-dispersible
BKC 0.05%	4.75 ± 0.23	1.99 ± 0.02	Aggregated

^1^ Weight per volume concentration in ACN. ^2^ Expressed as mean ± SD (*n* = 3). ^3^ Indicates the volume weighted diameter below which 50% of the total particle. ^4^ Calculated by dividing the difference between d_0.9_ and d_0.1_ by d_0.5_. d_0.9_ and d_0.1_ indicate the volume weighted diameters below which 90% and 10% of the total particle, respectively. ^5^ Visually evaluated after 24 h storage at room temperature. ^6^ Indicates without stabilizer.

**Table 2 pharmaceutics-11-00419-t002:** Compositions and physicochemical characteristics of TA microcrystals-loaded MSs.

	Compositions			Characteristics			
	TA crystal Size (µm) ^1, 5^	PLGA:PLA Ratio (w:w)	Drug:polymer Ratio (w:w)	Particle Size (µm) ^1, 5^	SPAN ^2, 5^	Loading Amount ^3, 5^	Loading Efficiency (%) ^4, 5^
F0	7.21 ± 1.02	1:3	1:5	15.9 ± 0.95	2.03 ± 0.04	0.16 ± 0.02	97.2 ± 2.35
F1	1.73 ± 0.02	4:0	1:5	16.2 ± 1.35	1.89 ± 0.05	0.15 ± 0.01	93.5 ± 0.87
F2	1.73 ± 0.02	3:1	1:5	17.2 ± 0.43	1.89 ± 0.02	0.15 ± 0.04	92.7 ± 5.76
F3	1.73 ± 0.02	2:2	1:5	16.1 ± 0.56	2.06 ± 0.02	0.14 ± 0.01	90.9 ± 1.01
F4	1.73 ± 0.02	1:3	1:5	16.9 ± 0.35	1.80 ± 0.01	0.16 ± 0.01	98.1 ± 1.84
F5	1.73 ± 0.02	0:4	1:5	15.9 ± 0.06	2.11 ± 0.01	0.15 ± 0.02	96.3 ± 2.48
F6	1.73 ± 0.02	1:3	1:2	18.9 ± 0.25	1.70 ± 0.01	0.31 ± 0.00	96.0 ± 0.18
F7	1.73 ± 0.02	1:3	1:3	15.8 ± 1.11	2.04 ± 0.07	0.22 ± 0.01	93.7 ± 1.33
F8	1.73 ± 0.02	1:3	1:10	16.9 ± 0.42	1.20 ± 0.01	0.09 ± 0.02	99.5 ± 2.55

^1^ Presented as d_0.5_ value; the volume weighted diameter below 50% of the total particle. ^2^ Calculated by dividing the difference between d_0.9_ and d_0.1_ by d_0.5_: d_0.9_ and d_0.1_ by d_0.5_: d_0.9_ and d_0.1_ are the volume weighted diameters below 90% and 10% of the total particle, respectively. ^3^ Calculated by dividing the weight of TA in microcrystals-loaded MSs by total weight of microcrystals-loaded MSs (mg). ^4^ Expressed as the percentage (%) after dividing the weight of TA loaded in MSs by total fed weight of TA (mg). ^5^ Expressed as mean ± SD (*n* = 3); Note: Lecithin was included in all formulations at the weight ratio of 5 *w/w* % to the total amount of polymers.

**Table 3 pharmaceutics-11-00419-t003:** Pharmacokinetic parameters of TA in plasma following IA administration of the marketed product, F4, and F8 in rats.

Parameters	Marketed Product	F4	F8
AUC_0–7day_ (ng·h/mL)	2787.4 ± 187.4	1500.4 ± 218.9 ^*^	1022.2 ± 161.9 ^*,**^
C_max_ (ng/mL)	218.7 ± 26.6	75.6 ± 17.2 ^*^	32.2 ± 12.0 ^*,**^
T_max_ (h)	3.7 ± 0.8	1.8 ± 0.4 ^*^	1.4 ± 0.5 ^*^
T_1/2_ (h) ^1^	5.3 ± 0.1	9.0 ± 0.5 ^*^	13.3 ± 2.7 ^*,**^

^1^ Calculated from the plasma concentration–time curve from T_max_ to 24 h after IA injection; Notes: Data are expressed as mean ± SD (*n* = 5). Statistical analysis was performed using the one-way ANOVA test; ^*^ significantly different from the marketed product (*p* < 0.05), ^**^ significantly different from F4 (*p* < 0.05); Abbreviations: AUC_0–7days_, area under the plasma concentration–time curve until day 7; C_max_, maximum plasma concentration; T_max_, time to reach maximum plasma concentration; T_1/2_, elimination half-life of the drug.

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
