# Peer review of "Design and In Vivo Pharmacokinetic Evaluation of Triamcinolone Acetonide Microcrystals-Loaded PLGA Microsphere for Increased Drug Retention in Knees after Intra-Articular Injection"

_pharmaceutics, 2019, doi:10.3390/pharmaceutics11080419_

Round 1

Reviewer 1 Report

The manuscript is original and interesting. The research is well-designed, aesthetically presented, the results are properly discussed and concluded. I recommend the acceptance of this manuscript for publication in current form with only a minor spell check.

Author Response

Thanks for reviewing the manuscript in detail: pharmaceutics-566876. The reviewer's comment was carefully studied and reflected in the revised manuscript. We tried our best to follow your precious commentary. Please find attached file, which address the issues raised by the reviewer, point-by-point.

Reviewer 2 Report

The manuscript give information about production and pharmacokinetic evaluation of loaded PLGA microspheres with increased drug retention in knees after intra-articular injection. This is a high quality work, well planned and discussed that deserves to be published in the journal.

I give some comments in order to clarify/increase the quality of the work.

In Fig#3, discussion the shape of release profile correspond to a mixture of diffusion+degradation(hydrolysis). I consider diffusion is more important than swelling. In fact the initial burst step correspond to external diffusion control, whereas from the slope (same) it would be possible to calculate the effective solubility of crystals in the media. Otherwise, the second step has a mixed control while as the final (more interesting for the slow release) step is consequence of polymer degradation.

Table #3: AUC units should be (ng) (h)/ml

From table 3 it is also remarkable that from F4 to F8 we change ratio 1:5 to 1:10 resulting loading 0.16 to 0.09 to produce AUC from 1500 to 1000. Area is related to the effective total quantity released, so F4 and F8 results are almost same (same bioalability) being the difference caused only for the different loading and proportional to that. I consider any comparison in this sense (for the same quantity injected in all cases and including marketed product) would be very interesting

Author Response

Thanks for reviewing the manuscript in detail: pharmaceutics-566876. The reviewer's comments were carefully studied and reflected in the revised manuscript. We tried our best to follow your precious commentary. Please find attached file, which address the issues raised by the reviewer. point-by-point.
